# Design for Manufacture and Assembly of Digital Fabrication and Additive Manufacturing in Construction: A Review

**Wiput Tuvayanond [1] and Lapyote Prasittisopin [2,\*]**

1   Faculty of Engineering, Rajamangala University of Thanyaburi, Pathum Thani 12110, Thailand
2   Faculty of Architecture, Chulalongkorn University, Wangmai, Pathumwan, Bangkok 10330, Thailand
\*   Correspondence: lapyote.p@chula.ac.th

**Abstract:** Design for manufacture and assembly (DfMA) in the architectural, engineering, and construction (AEC) industry is attracting the attention of designers, practitioners, and construction project stakeholders. Digital fabrication (Dfab) and design for additive manufacturing (DfAM) practices are found in current need of further research and development. The DfMA's conceptual function is to maximize the process efficiency of Dfab and AM building projects. This work reviewed 171 relevant research articles over the past few decades. The concepts and the fundamentals of DfMA in building and construction were explored. In addition, DfMA procedures for Dfab, DfAM, and AM assembly processes were discussed. Lastly, the current machine learning research on DfMA in construction was also highlighted. As Dfab and DFAM are innovated, practical DFMA techniques begin to develop to a great extent. Large research gaps in the DfMA for Dfab and DfAM can be filled in terms of integrating them with product structural performance, management, studied cases, building information modeling (BIM), and machine learning to increase operational efficiency and sustainable practices.

**Keywords:** design for manufacture and assembly; digital fabrication; additive manufacturing; 3D printing; construction; review





## 1. Introduction

In response to architectural and engineering needs for flexibility, complexity, high performance, intricacy, and customization of material/technology [1–4], the construction industry has to create novel technologies such as digital fabrication (Dfab) and additive manufacturing (AM). Although the construction industry has been identified as not only a large consumer of natural resources but also a big producer of environmental impacts, it is considered one of the inefficient manufacturing practices [5]. Automation in construction and architecture [6–8] was proposed as an alternative to costly and inefficient manufacturing practices. This digital architectural paradigm is anticipated to have favorable impacts on the built environments. As a result, the architectural profession is required to develop completely automated production forms and procedures that promote sustainability. Designers are an inevitable essential stakeholder in contributing to a greener construction due to their ability to design building activities such as material selection, site selection, transportation, construction method, building form, building envelope and facade, maintenance, and renovation of existing structures.

Understanding the influence of sophisticated technology on the field of architecture may direct future studies, inspire innovative design and construction techniques, and improve teaching strategies. AM technology is preferred above other Dfab technologies due to its operational potential in the architectural, engineering, and construction (AEC) sector. This approach might enable the sustainable construction of complicated building designs with less material and without the requirement for conventional formwork. AM technology may be utilized in all phases of the design process, from form-finding prototypes to the production of full-scale constructions.

AM is the process of printing multiple layers of materials on top of each other [9,10]. Frequently, the words "additive manufacturing", "rapid prototyping", and "3D printing" are used interchangeably to refer to the process of constructing an element through the progressive addition of material layers. ISO/ASTM 52900 [11] terms the AM as "a process of joining materials to make parts from 3D model data, usually layer upon layer, as opposed to subtractive and formative manufacturing methodologies." Since the mid-1980s, as Charles Hull invented the first commercial AM printer [12,13], this AM or 3D printing technology has been gradually evolving. Pegna [14] created the first large-scale concrete printer in the late 1990s, enabling the construction sector to adopt 3D printing. Although the creation of this technology began more than 30 years ago, its fast development began considerably later. The framework of new development showed that the number of articles on the use of 3D printing technology in the construction sector has risen over the past decade [15]. There is a rising interest in implementing and expanding this technology within the construction industry and, subsequently, throughout architecture. Recent architectural construction projects are worldwide built by a large-scale AM machine, and the AM instances of architectural buildings were displayed in Figure 1. The images were real construction projects gathered from open-access internet sources. The projects were built within the last five years in Dubai (UAE), Europe, the US, China, and Southeast Asia regions. It should be noted here that the AM construction can be built in several climate zones such as desert, tropical region, cold, and moderate temperature areas. This requires adjustment of concrete material to have proper characteristics for each climate.

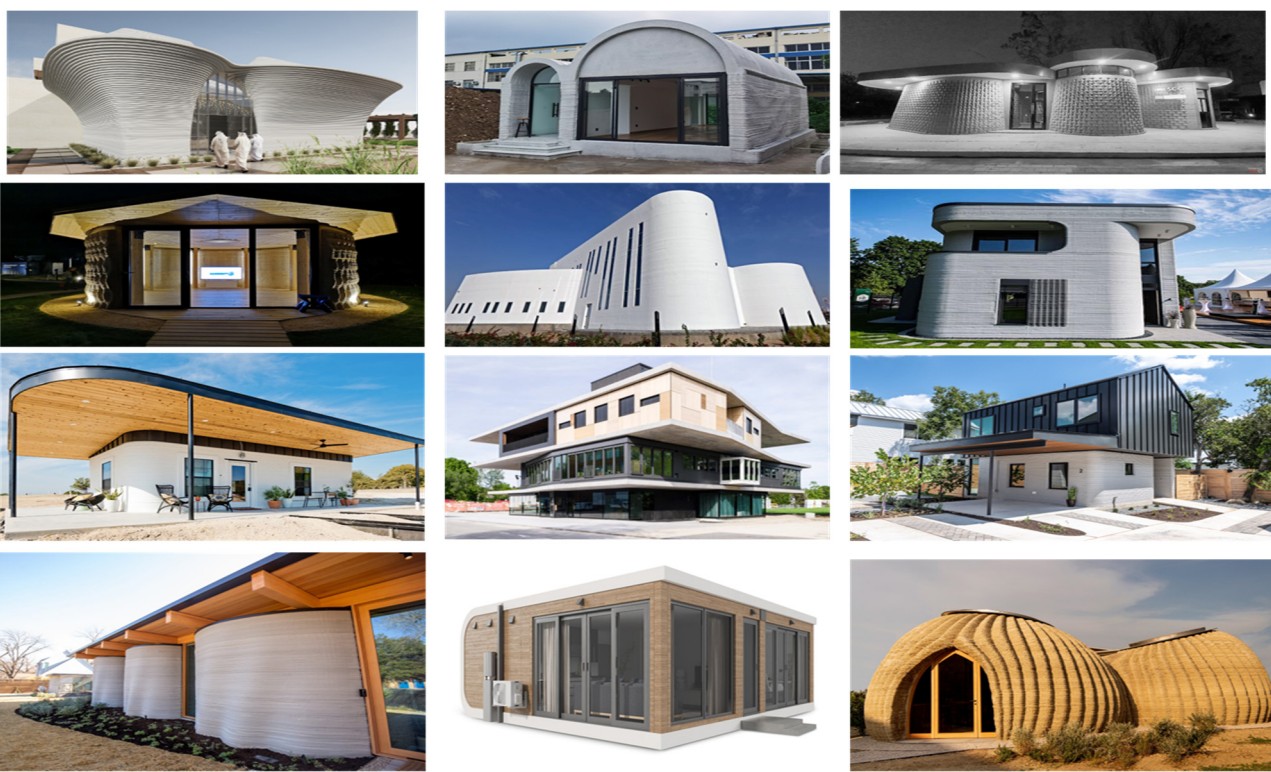

**Figure 1.** Recent instances of architectural AM projects.

Two methodologies of 3D printing for construction have been widely developed which are extrusion and powder-bed 3D printing cement. The concepts of both processes are exhibited in Figure 2. Numerous researchers have analyzed these technologies' influences and uses in the building industry [7,8,16–22]. The AM structures were mainly printed as concrete wall components. The AM wall can be designed as a load-bearing wall or a non-load-bearing wall depending on the concrete mix design. The concrete mixtures were adjusted to have good flowability, printability, and sufficient strength. In construction, the

size of AM wall panels is dependent on the printer's size. Several AM wall panels may be required for manufacturing and then fabricated into a larger building. Even though these studies are vast, they tend to concentrate on particular elements of technology and its application. However, while there are studies addressing various elements of AM technology, current research lacks the systematization required to offer a comprehensive overview of all the DfMA processes. It is found that AM construction can be well adopted by using current prefabrication techniques.

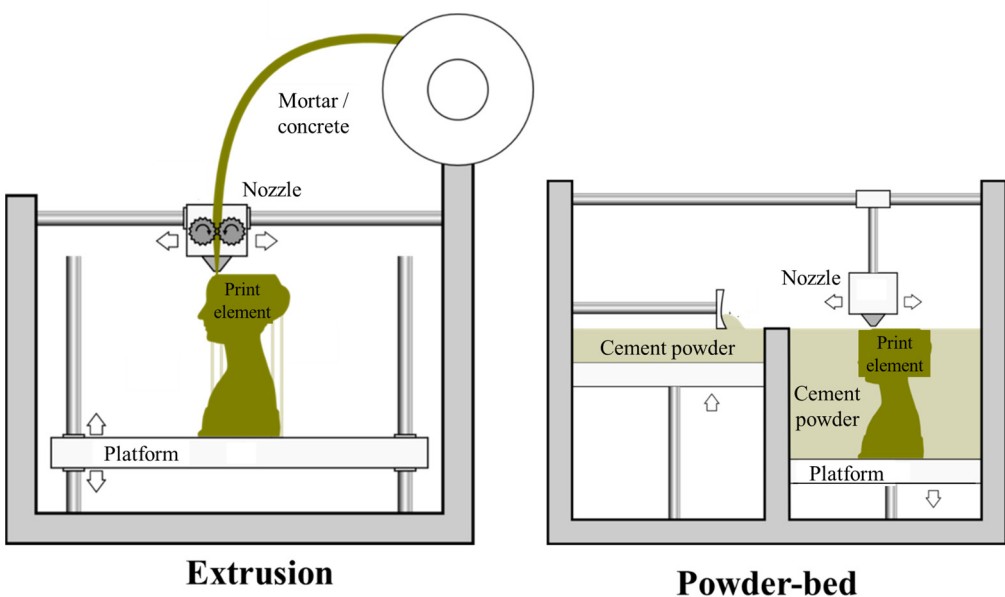

**Figure 2.** Concepts of extrusion and powder-bed 3D printing processes.

Prefabrication, sometimes known as offsite manufacturing, has been the subject of a large number of studies that have investigated many different practical aspects, including its business models [23–25], advantages and opportunities [26–28], and obstacles and restraints [29,30]. The DfMA technique is a set of methods for analyzing and enhancing product design for both economical production and assembly.

As the Dfab and AM technology is currently new and much effort has been directed towards research and development, the first emphasis for technologies' development is highly on material performance, effective construction, improved automation and machine friendliness to users, and implementation in a legitimate approach. Hence, very few studies sought to throw light on the best practices of design engineers, the manufacturing equivalents of architectural designers, in the design stage, such as the DfMA approach to the building [31]. The research gap can be apparently seen in this design area. This design procedure ultimately determines the overall building expenditure [32]. Undoubtedly, the question that DfMA will direct early in the process of product design efforts toward cost reduction. This will make DfMA feasible to reach the full lean production potential of the product since any potential manufacturing challenges and assembly concerns will already have been addressed in the design. This will make it possible to realize the entire lean production potential of the product. This reviewed study identifies 171 pertinent publications in the AEC sector that are related to DfMA, Dfab, and design for additive manufacturing (DfAM) practices. The PICO (population, intervention, comparison, and outcome) process was adapted as given in Table 1. It is noted that AM process involves design, printing, fabrication, transportation, and construction, while DfAM emphasizes the design stage for such AM technology. The article search is based on Scopus and Google Scholar databases. The structure of this reviewed paper begins with an introduction part, followed by the concept of DfMA, fundamental DfMA aspects in construction, DfMA for Dfab and DfAM, joints design for AM structure, and machine learning. Finally, implications, as well as conclusions and suggested future works, are addressed.

**Table 1.** PICO method of the review study.

| PICO Component | Explanation |
| --- | --- |
| Problem | Review current knowledge of DfMA for Dfab and AM process. |
| Intervention | Examine the effective methods and applications of DfMA for Dfab and AM process. |
| Comparison | Compare DfMA of prefabrication and precast construction. |
| Outcome | DfMA knowledge for Dfab and AM is currently under development and seemingly adopted similarly to prefabrication construction, and can be filled in terms of integrating them with product structural performance, management, studies case, BIM, and machine learning. |

As our current conventional methods are considered to be ineffective, Dfab and AM gain high attention to make the construction better. This review study primarily delivers effectiveness in Dfab and AM through design, such that innovative approaches (namely, design for effective uses and lean production) can be implemented throughout the design process and give efficiency gains and sustainable building and construction. This is a crucial step toward achieving AM's full potential.

## 2. Concept of DfMA

DfMA indicates an overall transition from a sequential, conventional approach to a non-linear, iterative design technique. Numerous DfMA processes and guidelines have been developed to assist designers in implementing this design philosophy to improve design, productivity, and profitability since its inception during World War II and growth extensively during the 1960s–1970s [33–38].

DfMA consists of two elements: (1) design for assembly (DfA) and (2) design for manufacture (DfM) [39]. DfA focuses on assembly, whereas DfM focuses mostly on the production of individual components [40]. During the 1980s, Boothroyd [41] and Swift et al. [42] developed the main principles of DfA and undertook a series of studies addressing assembly restrictions throughout the design phases. This aids in avoiding manufacturing and assembly problems in later phases of product development [43]. Based on the idea that the lowest assembly cost may be attained by creating a product that can be constructed economically using the best suitable assembly system. Stoll [44] mentioned that the important concept is to create a simplified design with fewer assemblages. The fewer components there are, the greater the likelihood that they will be correctly assembled. To accomplish this, Boothroyd [41] manually offered a variety of ratings for each component in the assembly process depending on the component's ease of handling and insertion. The well-established DfA principles are given in Table 2 (adapted from [45]).

The usage of DfA for AM with an emphasis on component decomposition, assembly-based re-design for AM, the decrease in assembly reorientation, and the number of parts through the development of an automatic DfA approach [46]. Robinson et al. [47] parameterized a DfA/DfM-based model. Using DfA and other design methodologies, El-Nounu et al. [39] redesigned a mechanical assembly using DfA. Furthermore, Manlig and Urban [48] analyzed the link between product development, material flow, and design life cycles for a specific product. In addition, a preliminary cost estimate of a hand pressure mop product was performed using both DfA and DfM [49]. Anyfantis et al. [50] designed multi-material mechanical components using both computer-aided DfA and DfM. Similarly, a strategy for cost-effective design was developed by Favi et al. [51].

DfM, on the other hand, evaluates the use of specified materials and manufacturing techniques for the assembly components, determines the cost impact of these materials and processes, and identifies the most effective design use [52]. DfM attempts to create parts that are simpler, less expensive, and more efficient to produce [43]. O'Driscoll [53] mentioned that DfM as the process of designing goods with manufacturing in mind had the objective of reducing manufacturing costs. Furthermore, the author asserted that the premise of DfM was at least 200 years old which was in the field of the handcrafted musket industry.

RIBA [54] advocated that DfM in construction was the process of planning such that specialized subcontractors could produce important design elements in the manufacturing framework. Panelized systems, such as claddings, have been created this way for years, and now the growing hybrid systems (i.e., unit pods), modular structures (i.e., completely factory-built homes), and 3D concrete printing also apply to the DfM principles.

**Table 2.** DfA principles (adapted from [45]).

| | Stage | Explanation |
|---|---|---|
| 1 | Functional analysis | Any material not qualifying for characteristics such as relative movement need and adjustment is excluded from the system. |
| 2 | Manufacturing process | Selection of materials, quantities, complexity, process, and cost for improved manufacturing. |
| 3 | Handling/feeding | A part's ease of manual or automatic assembly is evaluated (termed as feeding). |
| 4 | Assembly/jointing | Identifies and scores insertion, fastening, and gripping portions. This examination examines the ease of inserting and connecting pieces. Avoid fasteners. |
| 5 | Product group | A product's similar parts, assembly procedure, and routine feedings differentiate it from others. |
| 6 | Product structure | Structured information on manufacturing process description, materials selection, process variation for production, economics, design elements, size configurations, and process capabilities for tolerance and surface polish. |
| 7 | Component design | The designer is given information on insertion and fastening assembly processes, process capability data, component models, and assembly cost. |
| 8 | DfA heuristics | These are usually offered in pairs of "good practice" and "poor practice" examples. Graphically presented heuristic examples are simple to understand. |
| 9 | Evaluation assemblies | Two approaches to lower the overall number of components are presented, followed by a full investigation of fitting, handling/feeding, and fixing. Each component, part, and assembly procedure are scored to demonstrate complexity. |

From the aforementioned explanations of DfM and DfA, it is determined that these two disciplines should be viewed collectively as DfMA [55]. This is due to the fact that modern goods are complicated and the capacity to assemble them efficiently is equally essential. DfMA is a management and software solution that enables designers to address a product's material selection, design, and manufacturability at the outset [56]. Boothroyd [33] advocated the initial DfMA analysis technique, which established methodical processes for analyzing and enhancing product design for both cost-effective production and assembly. Ashley [52] stated that DfMA was strongly introduced in other high-tech industries such as aviation, it was labeled as a design review approach that determined the ideal part design, materials selection, assembly, and fabrication activities to generate a cost-effective product. The objective is to give manufacturing input in a logical and structured manner at the design conception phase.

## 3. Fundamental DfMA Aspects in Construction

Boothroyd (1994) advocated that DfA should be the primary concern for product design, resulting in a simplified product structure. Next comes the economical selection of materials and procedures, and then preliminary cost estimation. To reach a trade-off choice, cost estimates for the original design and the new (or improved) design will be compared thereafter. Once the materials and methods have been finalized, a more complete DfM study will be conducted. DfM is provided with standards, component design, and component assembly for lowering the total cost of production. The general series of DfMA procedures are illustrated in Figure 3 (adapted from [33]).

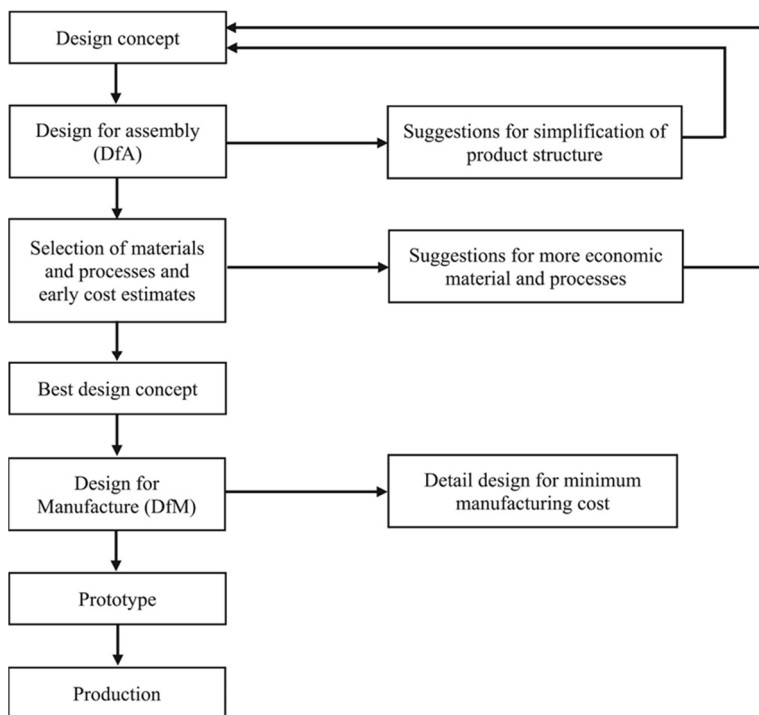

**Figure 3.** The general series of DfMA procedures (adapted from [33]).

Boothroyd et al. [57] enlisted the three major criteria for the application of DfMA to resultant products as shown below:

(1) The design team reduced the product's structure to save manufacturing and assembly expenses and to enhance product quality.
(2) A tool for quantities issues in their manufacture and assembly was developed.
(3) A tool for reducing costs and negotiating contracts with suppliers was also created.

Bogue [58] stated that there were three means to implement a DfMA procedure. One step was to adhere to a general set of qualitative and non-specific principles or standards and need stakeholders (usually designers and engineers) to interpret and apply them in specific cases. The objective was to include a variety of goods, techniques, and materials. The second technique quantifies the design. Each part's "assemblability" was scored. The last was a process automation technique where the design process might be quantified using computerization software. Similarly, Stoll [59] outlined ten DfMA principles and rules including (1) minimizing the total number of parts; (2) developing a modular design; (3) utilizing standard components; (4) designing parts to be multifunctional; (5) designing parts for multiple uses; (6) designing parts for ease of fabrication; (7) avoiding separate fasteners; (8) minimizing assembly directions; (9) maximizing compliance; and (10) minimizing handling. Kim et al. [60] also standardized 13 bridge constructions in the United Kingdom based on DfMA criteria. Jung and Yu [61] recently developed a DfMA checklist to evaluate the optima of design plans for offsite construction projects by outlining optimal design goals, the process, and DfMA principles. The documentation process of DfMA is still in the early stage.

Researchers and building owners are developing an interest in modular and prefabricated construction projects based on the DfMA. In these projects, building components are built in a factory and then sent to the construction sites, where they are assembled. Consequently, many research articles concentrating on the essential technologies for implementing DfMA in sustainable building, renovation, and interior projects were published [20,62–64]. For example, Serra [65] developed Australia's high-rise construction bathrooms with DfMA-based flat-pack walls saving almost one-third of operating energy usage owing to its efficient design. Furthermore, Wasim et al. [66] utilized DfA to quantify the efficiency of

prefabricated non-structural timber construction components for residentials. Their case study revealed that the DfA of the timber frame and drainage manufacturing system will be 9.8% and 10.244%, respectively. The DfMA can be performed for mechanical, electrical, and plumbing (MEP) systems for improving producibility and product quality throughout the product development process [37]. It is found that DfMA for modular and prefabrication techniques can be well applied to the Dfab and DFAM methods. The differences can be detailed such as differences in production and machine techniques, textures and patterns, structural performance, structural loading calculation, design softwares (i.e., G-code and slicing), and jointing techniques.

Exploration of industrial innovation, particularly offsite building, has presented DfMA with a distinct opportunity. DfMA is at the forefront of the industry's cross-sectoral learning and innovation agenda due to the parallels between offsite construction/prefabrication and manufacturing. In addition, rising technical innovations such as building information modeling (BIM) [67–69], 3D printing [4,70,71], the Internet of Things (IoTs) [72,73], and DfMA in particular, new entry opportunities of design and construction aspects for manufacturing expertise and efficiency improvement.

## 4. DfMA for Digital Fabrication (Dfab) and AM (DfAM)

In this section, two DfMA processes related to Dfab and DfAM were discussed. The number of technical publications that represent DfMA for Dfab and DfAM in construction is quite minimal. The authors reviewed existing articles from Scopus and Google Scholar databases relating to DfMA for Dfab and DfAM. Table 3 summarizes the existing 35 publications regarding DfMA for Dfab and DfAM. Based on these 35 publications, it is revealed much is emphasized on DfMA for DfAM (74%) in construction as illustrated in Figure 4a). The research publication analysis also found that current publications are published after 2018–2023 as shown in Figure 4b). Meaningly, the studies on the DfMA for DfAM topic are fairly novel and have been tremendously growing within the five recent years (about 70%).

**Table 3.** The summary of 35 existing publications on DfMA for Dfab and DfAM in construction.

| Year | Author | Process | Discussion | Reference |
|------|--------|---------|------------|-----------|
| 2011 | Williams et al. | DfAM | Design system focuses on three aspects: identifying essential use cases, defining formwork systems, and defining software element communication to facilitate expert user cooperation. | [74] |
| 2014 | Wang et al. | DfAM | Integration of 3D printing, BIM, and augmented reality is needed to improve architectural visualization in building life cycle. | [75] |
| 2015 | Bock and Linner | Dfab | Product structures and information aspects required manufacturing technology for full capability | [36] |
| 2015 | Yang and Zhao | DfAM | General Design Theory and Methodology (DTM) cannot take use of the enhanced design freedom and process options. Modifying standard DTM and DfAM can help designers effectively use AM in designs. | [76] |
| 2016 | Wu et al. | DfAM | BIM and 3D printing synergize to provide new DfMA possibilities in the building business. BIM can create an accurate 3D integrated information model for building design and 3D printing. | [5] |
| 2016 | Tang and Zhao | DfAM | Few product-level design approaches exist for both functionality and assembly, and some current design methods are challenging to execute due to an unfit CAD software. | [77] |

**Table 3.** *Cont.*

| Year | Author | Process | Discussion | Reference |
|---|---|---|---|---|
| 2016 | Tang et al. | DfAM | Establishes the basis for sustainable AM design through functionality integration and component consolidation. DfMA offers designs with fewer parts and less material without sacrificing functionality. | [78] |
| 2016 | Kim et al. | Dfab | An interview determines the acceptability of precast bridge components based on DfMA requirements. A case study on a newly completed highway bridge identifies the possibility of precast components selected from suitability analysis. | [60] |
| 2017 | Krimi et al. | DfAM | 3D printing provides design flexibility and cost savings to build complicated forms, not the time-saving. | [79] |
| 2018 | Arashpour et al. | DfAM | In advanced façade manufacturing, a substantial portion of the expenditure is for equipment such as CNC machines and 3D printers which can be significantly reduced by DfMA. | [80] |
| 2018 | Durakovic | DfAM | Most 3D printing studies are still in early stages. This method lacks numerous technologies; therefore, maturity will take time. | [81] |
| 2019 | Ng and Hall | Dfab | LEAN, DfMA, and Dfab share design to target value and concurrent engineering. | [82] |
| 2019 | Dorfler et al. | Dfab | Mesh mould is a novel construction technology for non-standard reinforced concrete buildings employing a mobile robot on site. | [83] |
| 2019 | Hinchy | DfAM | 3D printing is ideal for low-volume, sophisticated components, hence it should be selected over traditional methods. Build orientation and support structures affect manufacturing cost, time, post-processing, and final component mechanical characteristics. | [84] |
| 2019 | Medelling-Castillo and Zaragoza-Siqueiros | DfAM | Build orientation affects component stability during construction by determining the part's support surface on the building platform. | [85] |
| 2020 | Ng et al. | Dfab | Dfab manager and Dfab BIM coordinators are needed early in the design process. | [86] |
| 2020 | Alfaify et al. | DfAM | The suggested DfAM solutions include cellular structures, component consolidation and assembly, materials, support structures, build orientation, part complexity, and product sustainability. | [87] |
| 2022 | Nguyen et al. | DfAM | DfMA attempts to optimize product design to deal with complicated production processes while specifying 3D-printed product advantages throughout its consumption phases. | [88] |
| 2020 | Ghaffar et al. | DfAM | Collaboration across materials science, architecture/design, computer, and robotics is important to developing and implementing 3D printing. | [89] |
| 2021 | Gibson et al. | DfAM | Modern 3D printing has led to more emphasis on DfAM training. | [90] |
| 2020 | Frascio et al. | DfAM | This solution tackles the exponential link between construction volume and printer cost and improves efficiency by deploying many 3D printers simultaneously. | [91] |

**Table 3.** *Cont.*

| Year | Author | Process | Discussion | Reference |
|---|---|---|---|---|
| 2021 | Ng et al. | Dfab | Three design practices were identified: post-rationalization, mass customization, and modularization. | [92] |
| 2021 | Graser et al. | Dfab | Three theoretical factors for using Dfab house projects: full-scale projects are an effective Dfab strategy in AEC; large-scale implementation promotes Dfab's acceptability in AEC; and projects help develop a new Dfab paradigm. | [93] |
| 2021 | Ghiasian | DfAM | Intelligent machine learning-based recommender system that identifies part candidates and addresses AM infeasibilities in existing component designs. | [94] |
| 2022 | Prasittisopin et al. | DfAM | Small modules for 3D-printed pavilions can be attached together using bolt–nut designs. | [18] |
| 2021 | Morin and Kim | DfAM | The optimization scheme's effectiveness in breaking a cantilever beam structure into components that fulfill the AM build plate's geometric restrictions while reducing the structural impact of joints. | [95] |
| 2021 | Vu et al. | DfAM | DfMA framework entails three main elements: structure, property, and process. | [96] |
| 2022 | Ng et al. | Dfab | Proposed seven strategy propositions to achieve the benefits of adopting Dfab system. | [97] |
| 2022 | Rankohi et al. | DfAM | Integration of 3D printing, DfMA, and BIM can boost automation and productivity even with present labor difficulties. | [98] |
| 2022 | Sadakorn et al. | DfAM | Similar to the precast method, the jointing can be executed in dry process. | [99] |
| 2022 | Nguyen et al. | DfAM | Parametric model for bridge pier improved industrial output. | [100] |
| 2022 | Spuller | DfAM | Unlike product design application, construction occasionally uses DfAM. | [101] |
| 2022 | Song et al. | DfAM | New DfAM knowledge must be organized into general frameworks to assist practitioners throughout the product design process and to properly leverage present AM capabilities and developing potentials. | [102] |
| 2022 | Qin et al. | DfAM | Machine learning has contributed significantly to DfAM and has the potential to revolutionize AM. | [103] |
| 2023 | Rehman et al. | Dfab | Two most important liability factors are management capability and BIM. | [73] |

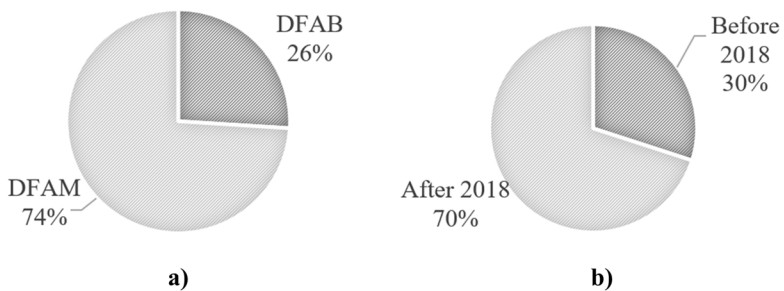

**Figure 4.** Relative research number of (**a**) DfMA for DfAM and Dfab in construction; and (**b**) published before 2018 and after 2018 (among a total of 35 existing papers).

*4.1. DfMA for Dfab*

Dfab is rising as a systematic breakthrough in the AEC sector to stimulate automation and enhance efficiency. It is necessary to incorporate manufacturing knowledge in the early design process. Bao et al. [63] addressed that a block-brick-based wall, hollow-brick-based wall, and shear wall system used skirting line connection, stitch connection, and tight connection, respectively. A paperless design and construction process can be supported by Dfab, which results in cost savings [104]. In addition, it offers a number of environmental, social, and economic advantages, including the reduction of waste, the removal of physical inventory, the reduction of labor, the implementation of digital quality control, and the establishment of an offline part setup [105]. The typical Dfab techniques consist of two methods computer numerical control (CNC) and laser cutting. Based on DfMA, Bridgewater [106] suggested DfA for factory-based production and onsite automation to reduce the number of components for Dfab such as robotics. He also mentioned rules for redesigning building systems for DfA, as well as a new construction contract and legal requirements for DfA. Bonwetsch [107] advocated that CNC let design information be sent directly and automatically to fabrication machines. Robotics focused on integrating design and construction, which helped to cut down on construction costs and time and improve the quality of design. Examples of how DfMA works for robotics and how codes and designs could be combined early in the design process were addressed.

The parameters found by Dfab could affect both design results and process. During the design process, all physical constraints of fabrication had to be taken into account. Martinez et al. [108] indicated how the robotized Field Factory System was designed using DfMA principles and how its production lines were set up. For instance, the factory layout took into account the size and range of motion of an ABB robot. The Service Core has been examined to improve the time and quality of assembly holistically. Montali et al. [109] determined the Knowledge-Based Engineering (KBE) approach using digital tools to support design through the automation of reusable knowledge on facade design with DfMA principles. They found that the 2D and 3D digital tools that were currently available could not close the design-manufacturability gap in the facade construction industry. The DfMA-based KBE for design automation was proposed to guide design from the beginning of the design process to improve quality, reduce delivery time and costs, cut down on rework, and support product development in construction. Furthermore, CNC milling was conducted to investigate the principles of DfMA [80]. Ng and Hall [92] conducted an online game with the Target Value Design (TVD) principle for modeling the Dfab construction. TVD principle implies a strategy that was built on lean principles and incorporates a design based on thorough cost estimates [110,111]. Concurrent engineering, design-to-target-values, and the maximization of values to project stakeholders were possibly conducted by TVD. They found that TVD was offered as a feasible design management strategy for managing Dfab during the design process and maximizing value for project stakeholders. However, the application of Dfab in TVD in the construction sector is still relatively new. The prerequisite for future assessment is required. Parametric modeling also supports collaborative work, which makes it easier to put DfMA into practice. Ng [97] reviewed 59 journal articles about Dfab and discussed how DfMA had several important enablers. These included Dfab engineers, parametric or computational resources, visual-programming conditions, bespoke/customized design and modular features, Dfab optimizing and prefabrication processes, an artifact of Dfab physical mockup, value of reducing human dependence, along with risks of increasing uncertainty in production and performance compromise/uncertainty. De Soto et al. [112] determined the productivity, cost, and time aspects of the onsite robotic fabrication technology. Results found that complex decoration structures could be made with Dfab at no extra cost. This is because Dfab can build a part in a more integrated way by obtaining feedback early in the design process, as also discussed in the full-scale Dfab house under the NEST project developed by EMPA, Switzerland [113]. Regardless of the fact that only a limited number of investigations have

been presently performed on Dfab technology, these Dfab principles are apparently in accordance with the DfMA principles and may be adopted without issue.

### 4.2. DfMA for DfAM

DfMA tools facilitate communication between product designers, production engineers, and any other stakeholders in the finished product. Barbosa [114] asserted that DfMA was an essential method for boosting the productivity of any product development via design in several manufacturing sectors. However, the AEC sector did not give building designers similar techniques. In an increasingly dispersed work environment, the integration of construction expertise into the design phases continued to rely on the experience of individuals [115]. Furthermore, Spuller [101] mentioned that in contrast to the domain of product design, the building sector made relatively infrequent use of these DfAM methodologies.

Figure 5 shows the complexity levels of DfAM techniques. Both direct component replacement and DfAM can be viewed as processes of manufacturing-driven and function-driven design strategies, respectively. The adaption of AM represents the medium ground between the two sides. To take advantage of AM, the design of a component can be modified, but its connections to other components are maintained in their previous states [116].

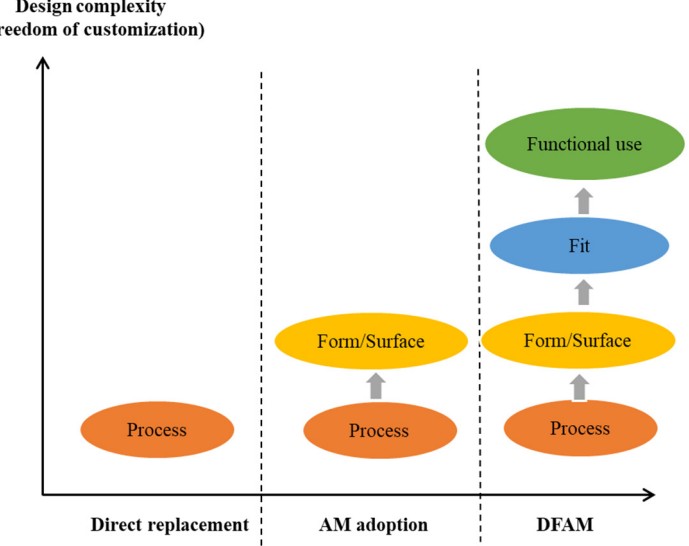

**Figure 5.** Complexity levels of DfAM.

First, the direct replacement (leftward) is the basic design process for the manufacturing process. From a traditional manufacturing standpoint, the *Handbook for Product Design for Manufacture* by Bralia [117] and *Product Design for Manufacture and Assembly* by Boothroyd et al. [57] addressed suitable instances of design for manufacturing standards and practices. The substantial work on design for manufacturing over many years indicated the complexity and pervasiveness of the design for manufacturing concerns [118]. It is necessary for designers to have a solid grasp of the limits imposed by accessible fabrication technologies. Some of these restrictions are alleviated by AM, while others are not. The applicability challenges for design for manufacturing in AM are shown in the following areas where traditional design for manufacturing falls short of the benefits offered by AM. The applicability challenges include:

- Layerwise operational characteristics and direct CAD model production extend part design creativity.
- Parts could be created as modular 3D puzzles incorporating small modules.
- As AM materials may be treated point-by-point or layer-by-layer, complicated material compositions and property gradients are possibly adopted.

- AM allows for the fabrication of hierarchically complicated, long-scale building designs.

AM's distinctive technique allows for low-cost, fast remanufacturing and repair. AM capabilities represent the complexity of shapes and surfaces in designs. It is feasible to create almost any form, allowing for various lot sizes starting from one, rapid customization of geometries, and shape optimization. Some studies determined using the inner truss as a surface of the architectural wall structure of the building [99,119]. Results indicated that several patterned AM wall structures could be created based on a geometric ratio. This led to a reduction in material consumption and printing time. Nguyen et al. [100] developed bridge constructions that were prefabricated using AM adoption. Throughout this work, a unique digital engineering model approach was developed by combining current knowledge of DfMA with structure-oriented parametric modeling technology. The geometrically complex elements of bridge piers that were aligned with the aesthetic surfaces were built using DfMA approaches and parametric modeling. The developed AM bridge pier is shown in Figure 6.

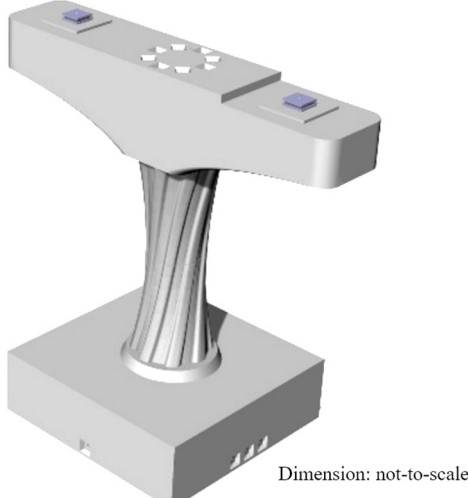

Dimension: not-to-scale

**Figure 6.** Digital modeling of prefabricated AM bridge pier [120]. Reproduced with permission.

Lastly, the DfAM as shown in the rightward of Figure 5 entails two additional steps (fit and functional use). The "fit" term means the assembly process. To reduce assembly time, cost, and challenges in conventional assembly, two primary ideas are frequently offered: reducing the number of pieces and eliminating fasteners. Both factors immediately result in fewer assembly procedures, which was the main cost driver [57]. Mavroidis [120] stated that, conventionally, the primary role of the assembly was to link together components, freeform material, and small elements to create a complex product. In contrast to typical assembly processes, AM permitted the consolidation of elements in locations where they were previously manufactured independently owing to manufacturing restrictions, material differences, and cost. AM reduces manufacturing limits and gives a fundamentally different viewpoint on jointing than conventional assembly. The issues associated with design considerations for AM assembly are covered as follows:

- The layer-by-layer or point-by-point nature of AM makes it easier to combine and embed parts. Most applications can be put into two groups: those that use operational mechanisms and those that use embedded components. In the case of operational mechanisms, if two or more parts need to be able to move in relation to each other, AM can build these parts already put together. For this type of non-assembly mechanism, one of the most crucial factors was joint clearance [121]. The joint clearance could reform the way the mechanism works. In addition, in the case of embedded compo-

nents, it is often essential in building a functional prototype by putting components into a part. This can improve the performance of the holistic system.

- AM is a good way to fabricate a structure with more than one material. The use of more than one material in AM can be applied to improve the functionality of the printed element. The multiple nozzle heads of extrusion AM have been examined [19,122,123]. Classen et al. [124] made fork-shaped, multi-nozzle extrusion heads for layer thicknesses of 50–100 mm and filament widths of 180–240 mm, as illustrated in Figure 7. The goal was to set up a fully automated, high-speed process for making continuously steel-reinforced concrete walls. Khoshnevis et al. [125] introduced supporting material, such as wax and sand, along with the concrete nozzle. This can be adopted for better buildability and can be built as a roof structure. Aside from these, multi-nozzle AM can produce complicated structures such as concrete extruded nozzles and spraying nozzles for either smoothing the surface or creating a range of surface textures.

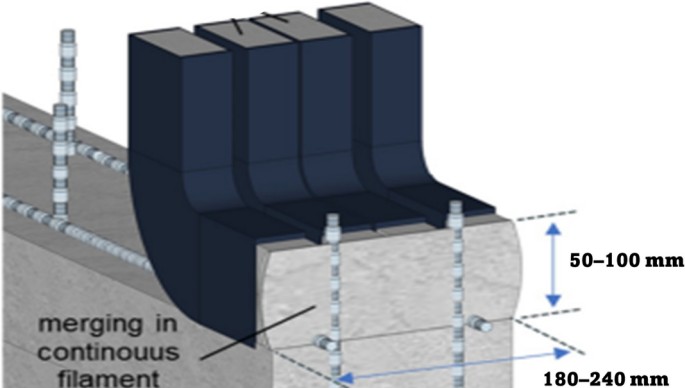

**Figure 7.** Multiple-nozzle print head for steel-reinforced concrete walls and two layers of steel reinforcement [124]. Reproduced with permission.

Another step, shown in Figure 5, is the "functional use" which can be mainly structural performance as a structural building component. Historically, products with basic geometries have been favored despite losing functionality or performance. This leads to material cost savings. To increase structural performance, AM structures are designed to be multifunctional and adaptable. The capability of DfAM to generate extremely flexible and functionally integrated components encourages the development of intelligent components that rapidly adapt to and respond to the operating environment [126,127]. Another virtue of AM is freeform printing, allowing for the creation of cellular structures. Based on topology optimization, it is possible to design a hollow structure that results in less weight and decreased material consumption. Nauyen and Vignat [128] asserted that the topology optimization approach permitted the identification of optimal material distribution and the reduction of material consumption while maintaining the mechanical qualities of the product. Additionally, in the case study of AM bridge piers, by relating the DfAM parameters to the estimated moment–curvature curves, the seismic performance of a bridge pier analyzed by the finite element method was achievable [119]. Vu et al. [96] advocated that optimized micro-structures could be self-supporting only in particular instances, such as when the load was equally distributed, and the micro-structures were anisotropic. Moreover, Morin and Kim [95] assessed the topology optimization for DfAM when the build area was limited. From their work, a structural cantilever beam case study was employed. Preliminary findings showed the optimization scheme's usefulness in decomposing the cantilever beam structure into components that could fulfill the AM build plate's geometric restrictions.

In addition to the structural performance aspect, other functional purposes such as thermal and acoustic insulation performance, MEP, and Heating Ventilation and Air Conditioning (HVAC) systems can be designed into the AM structure. Prasittisopin et al. [22]

developed a textured AM wall with a hollow structure that allowed the structure to perform thermal resistance to sunlight in a tropical climate. The AM wall could end up in electricity expenditure by almost 50%. Karadeniz and Toksoy [129] also mentioned that the HVAC system could be successfully implemented in AM through DfAM, followed by Heat Recovery Ventilation (HRVU) and Air Handling Unit (AHU) systems. DfAM methods were designed to aid designers in making decisions at the design stage to fulfill functional requirements while maintaining manufacturability in AM systems and to aid manufacturers in their fabrication process [85]. DfAM includes four steps for process, form/surface, assembly, and functional use, allowing for greater levels of design complexity or customization freedom.

Overall, 52 research papers were reviewed, and topics emphasized related to DfMA for Dfab and DfMA for DfAM were categorized. The relevant topics determined to entail product structure/performance, reference case, management (i.e., collaboration, training, and lean engineering), BIM, machine learning, CAD, modifying standard, and visualization. Figure 8 exhibits the research number of DfMA for DfAM and Dfab *relating to eight different themes. Existing research is still performed in the areas as followed: product structure/performance > management on collaboration, training, and lean engineering > adequate reference practices. Following the DfMA based on BIM, can result in the digitization of building models throughout the manufacturing and assembly operations. Few DfMA studies for construction have been conducted involved in machine learning, CAD, learning standard modification, and digital visualization technique such as virtual reality.*

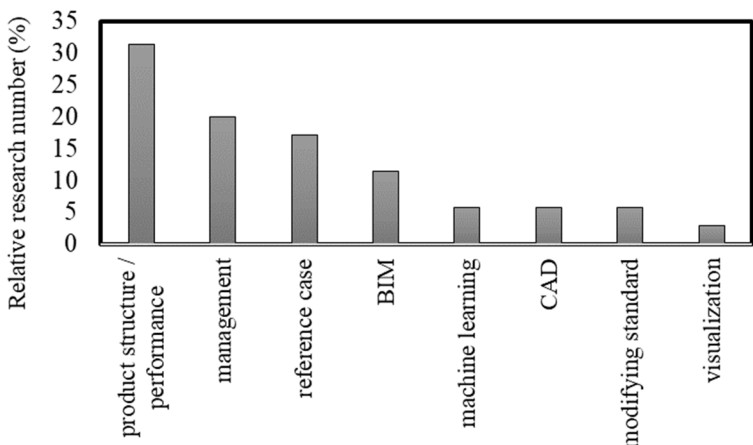

**Figure 8.** Relative research number of DfMA for DfAM and Dfab relating to various topics.

## 5. Joints Design for AM Structure

Some investigation programs determined the jointing process for prefabrication and cantilevered beam structure [18,95]. In the case study of DfAM for the cantilevered beam, the edges of the partitioning rectangles reflect the partitioning lines that divide the structure into components and can fit within an AM machine. To represent the structural impact of building a multicomponent system, joints are modeled at the dividing rectangle's borders. For optimization purposes, it is assumed that the joint material qualities are 15% weaker than the structural material properties. The decomposed design can be impacted by the joint design. This DfMA of jointing AM wall panels is not yet validated into basic practices. Some prototypes and idea concepts were dissimilated. The joint component is one of the most critical elements for buildings and structures because it relates to structural performance (both static and dynamic), acoustic and thermal insulation performance, and water/moisture leakages. The given case studies of DfMA for AM wall panels should be discussed.

For AM concrete pavilion, small modules were printed and then fabricated. Each module's joint assembly procedure consisted of two steps: (1) finding the connection location and (2) joining the small modules. The location for installing an anchor bolt at a joint is defined. Figure 9 depicts the locations of the joint regions and joint assembly

processes. First, the flat surfaces of each module were closely joined, and each module's height was split into five portions. Each part's height was dependent on the module's height, and two-thirds of each section was positioned in the joint area. It was proposed that the junction location be positioned roughly 150 mm within the outer shell to guarantee a secure connection between the two portions. It was proposed that the junction be secured using a 680 mm long (2.7 in long) anchor bolt. The angle of the anchor bolts was parallel to the shell's flat surface. Then, the joint system was built to connect each module with high precision and accuracy. Anchor bolts and studs were used to install each module. To build the assembly as planned, the piercing operation must be performed with precision. After the studs were inserted, knots were used to connect each module. All anchor bolts, studs, and knots were adapted from a stainless material, such as zinc-coated galvanized steel.

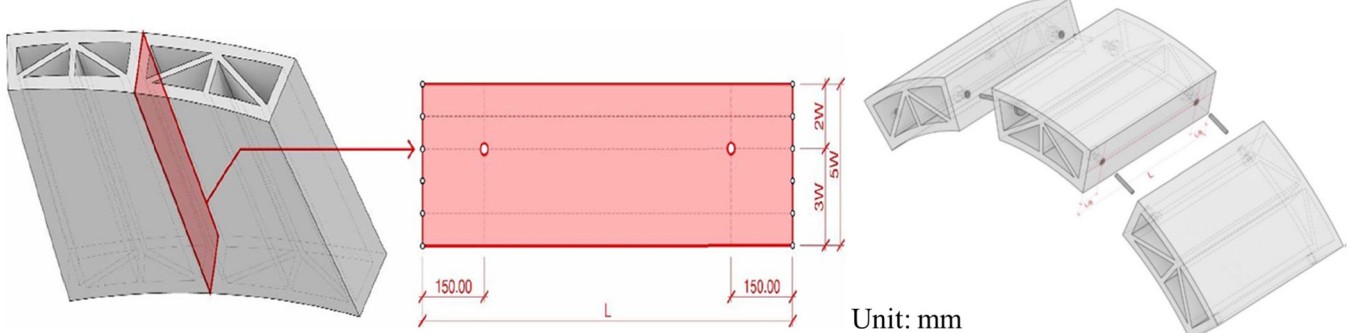

**Figure 9.** Location of the joint region and joint assembly processes of the 3D printing pavilion.

Another joint design of AM load-bearing wall structure incorporating concrete material by Sadakorn et al. [99] was mentioned. They proposed employing steel plates in dry production similar to precast wall parts. Figure 10 shows the precast wall element's steel plate and two bolts joint connection. Middle wall panels are where the bolts enter. The planned DfMA solution of AM wall panels is readily accessible. The suggested wall panel junction dimensions were also displayed. The AM load-bearing walls were jointed at both ends with projecting fins. The lift-up component must have an open hole that may be filled with cement and inserted in the lifting point. The horizontal wall joints were steel plates, $6.5 \times 12.5$ cm and 4 mm thick, with holes for tightening nuts to save installation time on site. The joint area was concave inward. The joints could be covered with cement plaster after installation to protect from leakages.

Frascio et al. [91] reviewed the jointing methods with adherents and adhesives. They discussed a variety of tailoring techniques for additively made adhesives, with the goal of optimizing the performance of bonded joints. Customizing AM adhesives according to the DfMA strategy has shown to be a very effective, although mostly unexploited, method for enhancing the performance of adhesively bonded joints.

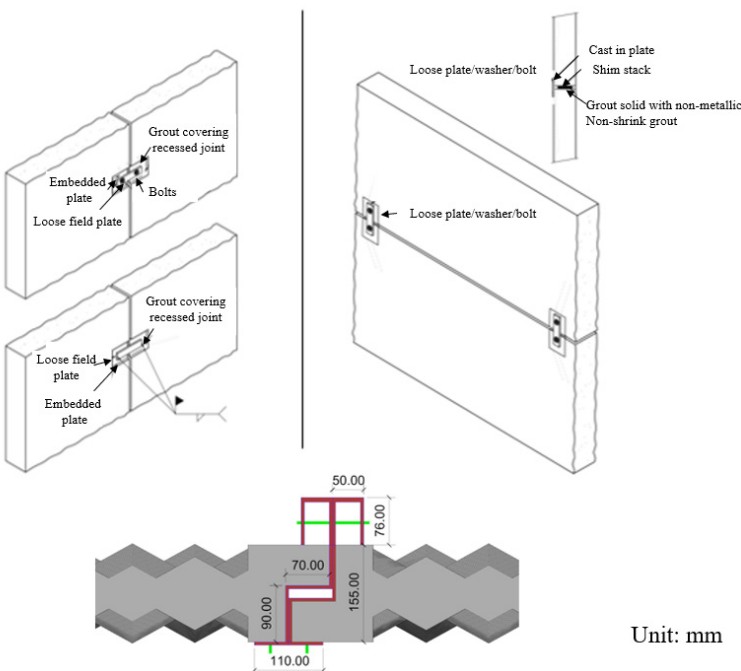

**Figure 10.** Dry joint design of AM load-bearing wall structure.

## 6. Machine Learning for DfAM

Machine learning is defined as "allowing computers to solve problems without being specifically programmed to do so" [130]. Due to the availability of vast amounts of data, the advancement of computer technology, and the improvement in the efficacy of accessible machine learning techniques, it is becoming a fast-emerging topic nowadays.

Several machine learning techniques in DfAM have been successfully developed in wide ranges of applications [131–134]. The main applications highly adopted by machine learning were those such as aerospace, automobile, and defense. These include multi-stage Bayesian surrogate models [135–137], artificial neuron network (ANN) [138–140], inductive design exploration method [141–143], support vector machine [69,144,145], graph convolutional networks [146], surfel convolutional neural network [147], multi-task Gaussian process learning algorithm [148], computational fluid dynamics model [149,150], back propagation neural network [151], and particle swarm optimization method [152,153].

Regarding the AEC industry, machine learning can be implemented effectively in any activity including a conceptual design phase, design optimization, cost prediction, transportation, and fabrication time. As et al. [154] proposed a graph-based machine learning system for 3D space, which was more organized and combinatorial than photos, text, or audio. They employed function-driven deep learning to develop conceptual design and then trained deep neural networks to evaluate existing graph-encoded designs, extract subgraphs, and integrate them. Yigit [155] and Huang et al. [156] used the machine learning method to optimize residential buildings for minimizing building energy consumption. This machine learning could train for passive design optimization of the green roof conducted by Lin et al. [157]. For predicting the construction cost, fabrication cost, total construction time, monitoring, and maintenance activities during the design phase, several works were evaluated both onsite and offsite manufacturing [158–162].

In terms of geometric flexibility and highly interconnected structures, AM had enabled novel product designs and performance improvements [163]. A digital database from AM can be adopted using machine learning techniques. The benefits of using machine learning can be implemented in several DfAM aspects. Machine learning can be beneficial in the following areas: (1) conceptual design phase, (2) design optimization, (3) geometry deviation prediction from build orientations and thermal deviations, (4) material analytics (such as material properties, material chemistry, material multi-structure, and resultant

performance), (5) prediction of defect in quality assurance process by image analysis, sensor signal methods, and (6) prediction of final product performance, total costs, energy consumption, and carbon emissions. Due to the unique production paradigm of AM, batch sizes, production schedules, and cost drivers may differ from those of conventional techniques. It also necessitates distinct methods of metrology and quality control. Therefore, DfAM has been presented as a means to provide AM design experts with a comprehensive set of design and analysis tools for complicated component structures and AM processes. Typically, DfAM consisted of two primary study topics: component design and design optimization [60]. AM offers free shapes and bespoke geometries for component design, enabling the production of intricate internal elements to boost functionality and improve the performance of target parts, providing designers with creative flexibility. AM component designers must define production route methods, part placements, build orientations, and support structures to improve the quality of final printed items for design optimization. Machine learning technologies have been increasingly utilized by DfAM in recent years [136] because of advancements in artificial intelligence, IoT, and data availability [101].

Very little machine learning research on the issue of DfAM for construction has been undertaken. Qin et al. [103] reviewed 222 latest research publications regarding machine learning for AM in several industries. However, only one paper was published based on using machine learning for DfAM with concrete material conducted by Lao et al. [164]. The researchers used an ANN model to establish a correlation between the nozzle and extrudate geometries. Upon completion of model development, a nozzle-extrudate database was created so that the ideal nozzle shape for a given extrude shape could be analyzed. Table 4 illustrates a summary of the process flow. During the pre-testing phase, the training data for the ANN model are compiled. After topology optimization, the predictive ANN model is then trained. By linking randomly produced nozzle geometries to their anticipated extrudate cross-sectional shapes, a database is created using the ANN model. Finally, nozzles for various target extrudate cross-sectional shapes may be retrieved from the database and employed in the printing process. The findings demonstrate that the suggested method enhances the surface quality of different structures with distinct contours.

**Table 4.** Summary of the workflow to identify nozzle shape using ANN model.

| Workflow | Discussion |
| --- | --- |
| Pre-testing | Set up nozzle experiments and perform experiments. |
| ANN model | Optimize topology, train, and validate. |
| Establish database | Generate sufficient volume randomly and predict extrudate shape. |
| Target extrudate cross-sectional shapes | Analyze target shape, find nozzle shape, and perform printing. |

A further recent publication on machine learning of DfAM in the object construction field conducted by Ko et al. [62] was present, even though it is not for the building. They employed a machine learning algorithm of Classification and Regression Tree on measurement data from the National Institute of Standards and Technology for the construction of a Laser Powder Bed design rule. Several construction members could be obtained using a machine learning algorithm including overhang, hole, beam, wall, cylinder, sphere, thin wall, and support structure. The material property could also be parameterized such as material distribution, material type, and thermal property. Many research programs can be extensively carried out on the machine learning of DfAM for the AEC industry such that the DfAM can be easily and effectively implemented.

## 7. Implications

The implementation of the DfMA of Dfab and DfAM technology can have positive effects on construction technology in three areas: economic, social, and environmental aspects. As previously noted, Dfab and AM technologies reduce labor expenses, which is

advantageous for the economy given that labor scarcity is one of the most significant global concerns. The labor scarcity causes increased labor expenses, which can be mitigated by automation technology during the preconstruction phase (i.e., design and manufacturing). Next, for the social aspect, it is found that construction can be risky for fatality compared to other businesses [158]. Hence, automation technology can also provide a more viable solution. Lastly, for environmental impacts, the positive implications can be delivered by both adopting low-carbon material selection and effective technology processes [165–170]. Adopting by-product waste into the AM material can be the key to lower carbon emissions to the industry as well as adopting such an effective process can decrease the wastes and formworks during fabrication. Last, the use of machine learning in DfMA can deliver a frontier innovation to design, train, and optimize with computerization. The frontier improves design prediction in many areas for designers including structure optimization, construction technology, 3D space, built environment and comfort, lighting, and energy efficiency of buildings.

DfMA of Dfab and DfAM is not only designed to cut production costs, but it may also construct buildings in remote and harsh temperature areas where it is challenging to transport construction materials, such as the North Pole and the desert. The objective is to construct utilizing resources that are native to the area. Typically, these are futuristic concepts for populating extraterrestrial worlds, such as constructing buildings from the lunar or Mars surfaces. Moreover, these structures must be self-sufficient and sustainable. For instance, the design study called the Mars habitat (MARSHA) of the multi-planetary architecture and technology design organization was awarded by NASA and this project was created by AI SpaceFactory [171]. The MARSHA habitat provides a view into the future of human existence on Mars, with a 15-foot-tall prototype 3D-printed building and three robotically placed windows. The MARSHA project was recognized for its innovative use of materials, which consisted of a biodegradable and recyclable basalt composite produced from Mars' native components. This composite material was stronger and more durable than its normal concrete-based materials.

Although without a doubt, Dfab and DfAM appear to be promising solutions for automated construction methods, developments in any domain are still ongoing. As of yet, none can validate the best practices for implementing them in actual construction projects. As seen today, a number of in-house prototypes have been publicized worldwide. A thorough understanding is not defined, leaving the question of whether the technologies can shift our current construction paradigms or merely fit with some sectors such as decorations and constructions in uncommon conditions.

## 8. Conclusions, Implications, and Suggested Future Works

A state-of-art review of the DfMA for Dfab and DfAM was performed to discuss the adoption in the AEC industry on various aspects, entailing the DfMA concept, DfMA implementation in construction, DfMA for Dfab and DfAM, joints design for AM assembly, and machine learning for DfAM. The key annotations from publications from the 1980s to recent developments were discussed as follows:

(1) AM using concrete materials also applies to the DfM and DfA principles suitably.
(2) Increasingly advanced technical developments in construction, such as AM and DfMA in particular, new entrances for manufacturing technology, and improvements in production efficiency.
(3) The majority of research (70%) has been investigated within this five-year period.
(4) DfAM allows for a greater degree of design complexity as well as a larger range of freedoms in terms of customization. It consists of four stages: process, form/surface, assembly, and functional usage.
(5) Existing knowledge is still applied to the product structure/performance, management, and BIM integration domains.
(6) Anchor bolt and stud fabrication is a viable option for achieving joint design in an AM wall structure. Additionally, the DfMA of AM wall structure can be designed in



a like manner to the precast wall system. More practices are required for validating these techniques.

(7)   Although many machine learning methods for DfAM have been studied in a variety of applications, only one or two research programs have been conducted in the building industry.

DfMA has lately been adopted in modern construction technologies such as prefabrication and offsite construction, and several future studies may be conducted in various facets including formal documentation, general case practices, and design process management. Regarding this review, it was apparently revealed that the DfMA for Dfab and DfAM is deficient since the new reference cases are still confined. It is possible to obtain the current DfMA for integration within design and construction, repair, renovation, and rehabilitation, leaving a large gap for researchers to fill so that the DfMA can provide significant advantages to the AEC sector. This is a crucial step towards realizing AM's full potential. Current trends of DfMA in Dfab and DfAM now also emphasize hybrid AM; several approaches, such as the mixing of different materials during a deposition under varying temperature conditions and applying reinforcement during printing can be employed to circumvent a number of restrictions. The hybrid process may be characterized as a method that combines many production operations from various manufacturing technologies. The merging of Dfab and AM technologies, for instance, can offer a novel method resulting in time and cost benefits.

**Author Contributions:** Conceptualization, L.P.; formal analysis, L.P.; writing—original draft preparation L.P.; writing—review and editing, W.T.; visualization, W.T.; funding acquisition, L.P. All authors have read and agreed to the published version of the manuscript.

**Funding:** This research was funded by the Multidisciplinary Research Grant, Faculty of Architecture, Chulalongkorn University and Thailand Science Research and Innovation Fund, Chulalongkorn University (SOC66250010).

**Institutional Review Board Statement:** Not applicable.

**Informed Consent Statement:** Not applicable.

**Data Availability Statement:** The data presented in this study are available on request from the corresponding author.

**Conflicts of Interest:** The authors declare no conflict of interest.

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
