# Peer review of "Design for Manufacture and Assembly of Digital Fabrication and Additive Manufacturing in Construction: A Review"

_buildings, doi:10.3390/buildings13020429_

Round 1
Reviewer 1 Report (New Reviewer)
The manuscript sounds interesting but the innovation of the paper must be highlighted in several areas of the manuscript. The manuscript should be revised keeping in view wider audience and not only target the technical aspects. Following revisions are require
1. Discuss with clarity how Dfab and DfAM can be filled with research gaps.
2. The current study's outcomes are unclear and should target wider area of audience.
3. The innovation must be highlighted at the end of the introduction part.
4. Discuss the disadvantages of conventional techniques to build a foundation for the said innovative part.
5. section 3 must be improved with more concise details.
6. Discuss the limitations of using the proposed methodology using Dfab and DfAM
7. The comparison of proposed methodologies with existing one must be presented with more clarity and detail
Author Response
The authors gratefully thank the reviewer for the in-depth reviews. This enhances the manuscript quality significantly.
- Discuss with clarity how Dfab and DfAM can be filled with research gaps.
Line 93-99: The authors added
“Because the Dfab and AM technology is currently new and much effort has done for research and development, the first emphasis for technologies’ development is highly on material performance, effective construction, improved automation and machine friendliness to users, and implementation in a legitimate approach. Hence, very few studies sought to throw light on best practices of design engineers, the manufacturing equivalents of architectural designers, in the design stage, such as the DfMA approach the building [31]. The research gap can be apparently seen in this design area.”
- The current study's outcomes are unclear and should target wider area of audience.
Table 1: The authors clarified the outcome of the study as:
“DfMA knowledge for Dfab and AM is currently under development and seemingly adopted similarly to prefabrication construction, and can be filled in terms of integration them with product structural performance, management, studies case, BIM, and machine learning”
- The innovation must be highlighted at the end of the introduction part.
Line:117: The authors highlighted the innovation by separating the paragraph and revised:
“such that innovative approaches (namely, design for effective uses and lean production) can be implemented throughout the design process…..”
- Discuss the disadvantages of conventional techniques to build a foundation for the said innovative part.
Line 115: The authors added “As our current conventional methods are considered to be ineffective, Dfab and AM gains high attentions to make the construction better.”
- section 3 must be improved with more concise details.
The authors concise contents in Section 3 as follows:
Boothroyd (1994) advocated that DfA should be the primary concern for product design, resulting in a simplified product structure. Next comes the economical selec-tion of materials and procedures, and then preliminary cost estimation. To reach a trade-off choice, cost estimates for the original design and the new (or improved) de-sign will be compared thereafter. Once the materials and methods have been finalized, a more complete DfM study will be conducted. DfM is provided with standards, com-ponent design, and component assembly for lowering the total cost of production. The general series of DfMA procedures are illustrated in Figure 3 (adapted from [33]).
Boothroyd et al. [57] enlisted the three major criteria for the application of DfMA to resultant products as shown below:
[1] The design team reduced the product's structure to save manufacturing and as-sembly expenses and to enhance product quality.
[2] A tool for quantities issues in their manufacture and assembly was developed.
[3] A tool for reducing costs and negotiating contracts with suppliers was also creat-ed.
Bogue [58] stated that there were three means to implement a DfMA procedure. One step was to adhere to a general set of qualitative and non-specific principles or stand-ards and need stakeholde (usually designers and engineers) to interpret and apply them in specific case. The objective was to include a variety of goods, techniques, and materials. The second technique quantifies the design. Each part's "assemblability" was scored. The last was a process automation technique where design process might be quantified using computerization software. Similarly, Stoll [59] outlined ten DfMA principles and rules including: (1) minimizing total number of parts; (2) developing a modular design; (3) utilizing standard components; (4) designing parts to be multi-functional; (5) designing parts for multiple uses; (6) designing parts for ease of fabrica-tion; (7) avoiding separate fasteners; (8) minimizing assembly directions; (9) maxim-izing compliance; and (10) minimizing handling. Kim et al. [60] also standardized 13 bridge constructions in the United Kingdom based on DfMA criteria. Jung and Yu [61] recently developed a DfMA checklist to evaluate the optima of design plans for offsite construction projects by outlining optimal design goals, the process, and DfMA princi-ples. The documentation process of DfMA is still in the early stage.
- Discuss the limitations of using the proposed methodology using Dfab and DfAM
In Section 7 (Line 599-605): The limitations are detailed as:
Although, without doubt, Dfab and DfAM appear to be promising solutions for automated construction methods, the developments in any domain are still on-going. As of yet, none can validate the best practices for implementing them in actual con-struction projects. As seen today, a number of in-house prototypes have been publi-cized worldwide. A thorough understanding is not defined, leaving the question of whether the technologies can shift our current construction paradigms or merely fit with some sectors like decorations and constructions in uncommon conditions.
- The comparison of proposed methodologies with existing one must be presented with more clarity and detail
Line 225-229: The comparison was clarified as:
It is found that DfMA for modular and prefabrication technique can be well applied to the Dfab and DFAM methods. The differences can be detailed such as differences of production and machine techniques, textures and patterns, structural performance, structural loading calculation, design softwares (i.e., G-code and slicing), and jointing techniques.

Reviewer 2 Report (Previous Reviewer 5)
The authors have added my comments. I agree to publish the manuscript.
Author Response
Author's Reply to the Review Report (Reviewer 2)
Thank you very much for your time and efforts.
Reviewer 3 Report (New Reviewer)
This paper presents the results of a review of innovative design and construction technologies in construction.
The review is extensive and well-documented, and the final summary is in line with the references.
I just have some minor remarks:
1. Please, avoid acronyms in the abstract and title and introduce the acronym for “Design for Manufacture and Assembly” (DfMA) in the introduction instead.
2. The English needs to be improved; the syntax structure is often poor.
3. Fix the references’ numbers in Table 3.
4. Is it necessary to have section 5 “Joints design for AM structure”? The review paper seems to focus on the overall characteristics, performances, and advances of digital fabrication and AM technologies; thus, a sudden focus on a specific topic may appear out of context.
Author Response
Thank you for your time and efforts. The authors appreciated these valuable comments.
- Please, avoid acronyms in the abstract and title and introduce the acronym for “Design for Manufacture and Assembly” (DfMA) in the introduction instead.
The acronyms were removed.
- The English needs to be improved; the syntax structure is often poor.
The manuscript was revised and proof-readed.
- Fix the references’ numbers in Table 3.
Thank you very much. The numbers were corrected.
- Is it necessary to have section 5 “Joints design for AM structure”? The review paper seems to focus on the overall characteristics, performances, and advances of digital fabrication and AM technologies; thus, a sudden focus on a specific topic may appear out of context.
The authors considered the more specific of “assembly” or DfA method in this section. None practices of joint design is publicized before so it should be valuable to readers. The authors kindly still present this section.

Round 2
Reviewer 1 Report (New Reviewer)
The changes have been done satisfactorily.
This manuscript is a resubmission of an earlier submission. The following is a list of the peer review reports and author responses from that submission.
Round 1
Reviewer 1 Report
The article looks good to me. However, the following suggestion would contribute to the article's quality.
1. Add the organization of the article.
2. Add the implications of the article.
3. If possible, please raise the quality of some figures. For example, in figures 11 and 2 some values/words are unclear.
4. Please align the center for some figures. For example, the position for figures 4 and 5 are not the same as 7.
5. I cannot see the methodology process for identifying 153 pertinent 84 publications in the AEC sector that are related to DfMA, Dfab, and DfAM practices.
Author Response
Dear reviewer,
Please kindly see attached the response file.

Reviewer 2 Report
Dear authors,
Thanks for contributing your work to this journal with “Design for Manufacture and Assembly (DfMA) of Digital Fabrication (Dfab) and Additive Manufacturing (AM) in Construction: A Review.” However, major work has to be done before this work can be published.
1) Please improve the English. Some sentences are missing a verb or preposition, which makes them difficult to understand or unintelligible:
· Line 13: Digital fabrication (Dfab) and design for additive manufacturing (DfAM) practices are found in apparent need for development
· Line 28-29: Although the construction industry has been identified as a big consumer of resources and has a substantial environmental impact
· Etc.
In general, there are many grammar mistakes that often makes the sentences difficult to understand or even misunderstand:
· Line 43: either "through to" or simply "to"
· Line 62: It is either "these technologies" or "this technology"
· Etc.
2) Ensure that there is consistency in verb tenses throughout the article (line 59 are instead of were, etc).
3) The text should be justified. Also pay attention to spacing and line spacing.
4) There are some repeated sentences and words: Lines 81-110, line 290,
5) Line 34-35: “equality, sustainability, democracy, diversity, and inclusivity.” This is not related to anything else being discussed, nor is it backed up by anything else. What is the relevance?
6) Line 60: Is the material used in construction really melted? if not, using the term FDM is not accurate.
7) Figure 2: If, indeed, FDM does not apply to this case, the figure should be changed.
8) Line 98: Just a suggestion, when Boothroyd is mentioned, especially when it is related to DfA, it is usually accompanied by Dewhurst, as they both developed the best known DfA method.
9) Line 166: You say there are three means to implement DfMA but the only explain one step.
10) Line 185: One third of what? Costs? Materials?
11) Line 189: MEP acronyms not previously explained.
12) Line 192: Which opportunity? For what?
13) Line 201: Who determines this? Quotation needed.
14) Line 202: Which relevant research publications? Quotation needed.
15) Line 248: TBD is TVD
16) Line 342-343: The sentences are confusing.
17) Line 349: Can be built as the roof structure or can build the roof structure?
18) Line 358: Explain why they have been favoured.
19) Line 379: HVAC acronyms not previously explained.
20) Line 384: HRVU and AHU acronyms not previously explained.
21) Line 390: How many were researched?
22) Lines 395-397: ???
23) Line 403 (and the following): Joints design
24) The whole of point 5 seems unrelated to the topic at hand as there is no explanation of DfMA or Dfab; it is only an explanation of two designs that have been carried out and no clear mention of how DfMA strategies have helped to achieve those designs.
25) Line 407: Which one? or is it "within an AM machine"?
26) Line 464: How can they be implemented?
27) Line 509: What is the basis for this assertion, or its relevance?
28) Lines 512-515: This is more of an introduction than a conclusion
29) Conclusions 3 and 4 are the same one but differently worded (also, check for grammar mistakes)
30) Line 529: This conclusion is only related to AM, it has nothing to do with DfMA (or the relation is not explained)
Please, all the pictures must be labelled with the correct copyright text.
Author Response

(The authors gave the same response as above.)

Reviewer 3 Report
This paper reviews the design for manufacture and assembly (DfMA) of digital fabrication (Dfab) and additive manufacturing (AM) in Construction. It first introduces concept of DfMA and fundamental DfMA aspects in construction, and then introduces the DfMA for digital fabrication (Dfab) and AM (DfAM), which is further followed by the jointing design for AM structure and machine learning for DfAM. I think the logic of expression is reasonable and clear.
Comments:
1. reference are not relevant. For example, line 460, ref. 142, this one has nothing to do with additive manufacturing, or DfAM. There are also some other ones like this. Need to check thoroughly. In addition, it would be more helpful to extend discussion and add some relevant references in section 6 regarding how machine learning can be used in areas such as conceptual design phase, design optimization, etc
2. Figures need to be improved. For example, Figure 7, 8, 10. Also, Figure 4, 5 may be unnecessary as they are just a ratio, or at least, put them together.
3. (Major) As a review paper, the key point is the perspective of the field, not just summary of the previous work. I only found a few perspective discussions in the last a few lines in Section 7. I don't think the depth of the authors' perspective qualifies a review paper with good quality. I suggest the authors to extend their discussion and come out with deeper and unique viewpoint.
4. Figure 1 refers to the large scale AM machine, a more detailed description such as what a large scale AM machine is (compared with normal AM machine), and how it prints large-scale construction would be interesting and required for readers who knows AM but not working on construction field, or readers who knows construction but not knowing AM enough.
5. line 306, "AM's distinctive technique allows for low cost, fast remanufacturing and repair." seems to need an indent. Please check thoroughly in the paper for misspelling and typesetting.
Author Response

(The authors gave the same response as above.)

Reviewer 4 Report
-A systematic review of the literature (RSL) is necessary to report a paper review. I recommend implementing PICO (population, intervention, comparison, and outcome) methodology or PRISMA methodology. Please do one of the two to improve the paper review.
-Making bibliometric analyses is an important activity of RSL.This software could help with the mentioned activity:
VOSviewer.
https://www.vosviewer.com/
and a general tutorial: https://www.youtube.com/watch?v=dsIKiABKdyI
-In Figure 2, it is necessary to explain how powder-bed 3D printing processes work in ceramic materials.
Lines 203-206
The research publication analysis also found that current publications were published after 2018–2023 as shown in Figure 5. Meaningly, the studies on the DfMA for DfAM topic were 205 fairly novel and has been tremendously growing within the five recent years (about 80%).
Is 74% or 80%? Please clarify.
Line 218-219
“It is necessary to incorporate knowledge about the manufacturing process at an early point in the design process.”
Please provide some concrete examples to back up this assertion. It is important to explain the brick wall concept between design and manufacture.
Lines 286-287
-In Figure 6. Complexity levels of DfAM, please put in upper letter the term in the bubble and in different color (coul be black).
Lines 429 and 449.
-Figure 10. Location of the joint region and joint assembly processes of 3D printing pavilion. Please specify the unit system (MKS or SI- UK System) mm or inches. Same topic for figure 11.
Please fix and standardize the unit system throughout the all text.
Line 522
“The majority of research, which accounts for 80 percent, has been investigated within 522 these five years”
Please verify this conclusion.
Please consult the following concepts in production terms: DFMA, DFM, and collaborative engineering.
I recommend that you use or include this literature in your paper review:
-L. Frizziero, G. Donnici, A. Liverani, and K. Dhaimini, "Design for additive manufacturing and advanced development methods applied to an innovative multifunctional fan," Int. J. Manuf. Mater. Mech. Eng. , vol. 9, no. 12, pp. 1-32, 2019, doi: 10.4018/IJMMME.2019040101
-P. Yuliarty and H. Ardiwijayanta, "The design of front and back grille of KAD-927 B fan with Nigel Cross Approach at PT. X (A Manufacturer of Household Appliances)," in IOP Conference Series: Materials Science and Engineering, 2018, pp. 1-8, doi: 10.1088/1757-899X/453/1/012035
-G. Donnici et al. , "A new car concept developed with stylistic design engineering (SDE)," Inventions, vol. 5, no. 3, pp. 1-22, 2020, doi: 10.3390/inventions5030030.
-J. EL Mesbahi, I. Buj-Corral, and A. EL Mesbahi, "Use of the QFD method to redesign a new extrusion system for a printing machine for ceramics," Int. J. Adv. Manuf. Technol. , vol. 111, pp. 227-242, 2020, doi: https://doi.org/10.21203/rs.3.rs-265668/v1.
-R. Jacobs and R. B. Chase, Operations Management. Production and Supply Chain, 15th ed. Mexico: McGraw-Hill Interamericana, 2018
-S. Kalpakjian and S. Schmid, Manufactura, Ingeniería y Tecnología. 2008
Author Response
Dear Reviewer,
Please kindly see the attachment.

Reviewer 5 Report
The authors submitted the manuscript Design for Manufacture and Assembly (DfMA) of Digital Fabrication (Dfab) and Additive Manufacturing (AM) in Construction: A Review.
I miss the point and purpose of contributing to an impacted journal. Describing a literature review from past to present and including in the objectives the information that it is a review of the literature over the last 5 years is not sufficient.
On the face of it, the paper is interesting and will certainly find a professional readership. However, the manuscript requires major revision and a few minor revisions.
Major revision:
· Are the examples in Figure 1 real or are they visualizations? At first glance, it seems to me that these are visualizations. I lack relevant sources where these photos/visualizations are from.
· I lack a comparison of the visualizations, how the buildings differ (materials, time of production, location (North Pole, desert, e.g. Dubai where the average annual temperature is over 36°C, civilized area) and add information about sustainability.
· Emphasize what the aim of additive technologies in construction is. It is not only to reduce production costs. It is important to construct buildings in remote locations where it is difficult to transport building materials. The goal is to build with materials that are in that location. Typically, these are futuristic ideas to populate alien planets, where the idea is to build buildings, for example, from the lunar surface. At the same time, these buildings must be self-sufficient and sustainable.
· It would certainly be useful to write information about futuristic themes in the review. For example, the ambitious Space Factory 3D Printed Mars Habitat project. https://www.cnet.com/pictures/this-3d-printed-mars-habitat-could-be-your-new-home-in-space-marsha-ai-spacefactory/null/. The project seeks to build sustainable buildings on Mars. It will be difficult to transport building materials there. Outline what the problems associated with this are (unfamiliarity with building materials, etc.).
Minor revision:
· The authors describe the distribution of AM by technology. In which category would you put DfAM?
· Add scale or pillar dimensions to Figure 7.
· Weld joint details (dimensions and text) in Figure 11 above are not legible.
· What is the difference between 3 and 4 in the conclusion: 3. Most research (80%) has been investigated within these 5 years / 4. The majority of research, which accounts for 80 percent, has been investigated within these five years.
I recommend the authors do:
· a clear review of which journals deal with this issue.
· Add current trends and use of AM in construction,
· Describe the studies in Figure 1 in more detail and describe how the case studies are different,
· Finally, describe the future and technological challenges and see reference to SpaceFactory and similar projects.
The manuscript requires considerable proofreading and revision.
After incorporating comments and re-reading, I will consider publishing the manuscript in the journal Buildings.
Author Response

(The authors gave the same response as above.)

Round 2
Reviewer 5 Report
The authors have added my comments. I agree to publish the manuscript.